# Towards General Agentic Intelligence via Environment Scaling

## Abstract

Advanced agentic intelligence is a prerequisite for deploying Large Language Models in practical, real-world applications. Diverse real-world APIs demand precise, robust function-calling intelligence, which needs agents to develop these capabilities through interaction in varied *environments*. The breadth of function-calling competence is closely tied to the diversity of environments in which agents are trained. In this work, we scale up environments as a step towards advancing general agentic intelligence. This gives rise to two central challenges: *(i)* how to scale environments in a principled manner, and *(ii)* how to effectively train agentic capabilities from experiences derived through interactions with these environments. To address these, we design a scalable framework that automatically constructs heterogeneous environments that are fully simulated, systematically broadening the space of function-calling scenarios. We further adapt a two-phase agent fine-tuning strategy: first endowing agents with fundamental agentic capabilities, then specializing them for domain-specific contexts. Extensive experiments on agentic benchmarks, $\tau$-bench, $\tau^2$-Bench, and ACEBench, demonstrate that our trained model, **AgentScaler**, significantly enhances the models' function-calling capability.

## 1 Introduction

Function calling empowers language agents to interface with the real world (Qin et al., 2023; Chen et al., 2024b; Qin et al., 2024; Schick et al., 2023; Su et al., 2025b). Yet, their progress is fundamentally constrained by the scarcity of agentic data[1], *i.e.*, trajectories generated by autonomous agents interacting with environments via explicit action executions, namely, tool calls (Zhou et al., 2023; Liu et al., 2024a). The community has gradually transitioned from the era of raw corpora and human-curated data to the emerging **era of experience** (Silver & Sutton, 2025; Wu et al., 2025a; Li et al., 2025; Tao et al., 2025; Geng et al., 2025). Crucially, language agents must experience these interactions themselves in a predefined environment, which makes both data collection and reliable supervision highly challenging.

Several approaches have been attempted to generate synthetic agentic data. Broadly, previous methods fall into two categories. The first category follows a reverse paradigm, in which user queries are generated to match each assistant function call observed at every interaction turn (Yin et al., 2025), though the resulting trajectories may exhibit limited realism. The second category follows a forward paradigm, which we refer to as simulated agent–human interplay (Chen et al., 2024a; Liu et al., 2024b; Prabhakar et al., 2025a; Barres et al., 2025; Zeng et al., 2025). Such generated trajectories, however, may lack naturalness. In this category, a high-level user intent is first formulated to necessitate agent interaction. Agentic data is then constructed in a top-down manner based on this intent through human–agent interplay. Yet, the environment is not scalable: the absence of automated environment construction hinders large-scale deployment and inevitably entails some degree of manual intervention.

To address these challenges, we pursue the advancement of general agentic intelligence via systematic environment scaling. Our approach follows a principled two-stage pipeline: *(i) **fully simulated environment construction and scaling***, responsible for establishing and expanding diverse agentic

---

[1]In this paper, the terms "function-calling", "tool", "API", "MCP" are used interchangeably; "agentic data" refers to trajectories involving such interactions.

scenarios, and *(ii) **agent experience learning***, which exploits these environments to foster generalizable intelligence.

In designing environment construction and scaling, we follow the principle that the core of an agent lies in its capacity for environment interaction, with each environment instantiated as a `read-write` database (Barres et al., 2025; Zeng et al., 2025). Specifically, we collect a broad spectrum of APIs and organize them into domains using community detection, where each domain represents an environment aligned with a specific database structure. Then, we instantiate tools as executable code, thereby achieving programmatic materialization that enables direct operations on the underlying database structures. Finally, we sample from the domain-specific tool graph to generate parameters for the tool sequences and initialize the corresponding database state. We then integrate these components into an overall user intent, grounding tool executions directly on the database. This design enables verifiability at both the environment level and the tool-argument response level.

For learning from agent experience, our focus is on training the agent's ability to perform tool calls and to respond effectively to users (Ye et al., 2025b; Su et al., 2025a). We begin by performing simulated human–agent interactions on the constructed agentic tasks (Prabhakar et al., 2025a), thereby collecting trajectories that serve as the agent's experience and perform strict filtering. To facilitate the acquisition of this capability, we adopt a two-stage agent experience learning framework: in stage 1, the agent acquires fundamental tool-calling skills across general domains; in stage 2, it is further trained within target vertical domains using domain-specific scenarios, enabling smoother and more context-aligned development of agentic capabilities.

Extensive experiments on agentic benchmarks, $\tau$-bench (Yao et al., 2024), $\tau^2$-Bench (Barres et al., 2025), and ACEBench (Chen et al., 2025) show the effectiveness of our pipeline and trained models. Based on the above pipeline, we train our family of **AgentScaler** models (4B, 8B, 30B-A3B), built upon the Qwen-3 (Team, 2025b) series. At each comparable scale (4B, 8B), our models achieve state-of-the-art performance. Notably, AgentScaler-30-A3B sets a new state-of-the-art with significantly fewer parameters among models with comparable active parameter size, delivering results on par with existing 1T-parameter models and leading closed-source systems. We also provide a systematic analysis covering model generalization, stability, and the long-horizon tool-calling challenge, offering key insights into the development of general agentic intelligence.

## 2 ENVIRONMENT BUILD AND SCALING

**Design Principal** There already exist many real-world environments for agent interaction (Qin et al., 2023). However, these real environments suffer from several practical limitations that make them unsuitable for large-scale and stable training. First, many real online APIs are unstable, frequently affected by rate limits, fluctuating QPS capacity, and transient API failures. Such instability prevents agents from receiving consistent tool feedback, which is crucial for effective learning. This limitation has also been highlighted by multiple recent works (Guo et al., 2024; 2025), motivating the development of simulated environments to enable reliable and large-scale agent training. Second, simulated environments can provide deterministic, reproducible, and controllable tool feedback, which real-world environments cannot guarantee. This stability is especially important for training models that rely on precise multi-step tool interactions (Sun et al., 2025; Su et al., 2025b). In essence, any function call can be interpreted as a `read-write` operation over an underlying environmental database $\mathcal{D}$ (Guo et al., 2025). Specifically, each function $func$ can be assigned an operator type, $\text{op}(func) \in \{\text{read}, \text{write}\}$, where `read`-type function perform queries over $\mathcal{D}$ (*e.g.*, retrieval, inspection, monitoring), while `write`-type tools induce state transitions in $\mathcal{D}$ (*e.g.*, modification, generation, actuation). Under this abstraction, a tool response is equivalent to evaluating the induced operator on $\mathcal{D}$, *i.e.*, $\text{API}(func, \alpha) \equiv \text{op}(func)(\alpha; \mathcal{D})$, where the symbol $\alpha$ denotes the input arguments provided to a function call. Furthermore, let $\mathcal{T}_d$ denote the set of tools within domain $d$. Tools in the same domain typically exhibit structurally similar read–write patterns, which can be captured by a common database schema $\mathcal{S}_k$. Consequently, the design problem reduces to defining a partition of the tool space into domains $\{\mathcal{T}_1 \ldots, \mathcal{T}_M\}$, and assigning to each domain a database schema $\mathcal{S}_k$, where $\mathcal{S}_k$ specifies the environment for that domain.

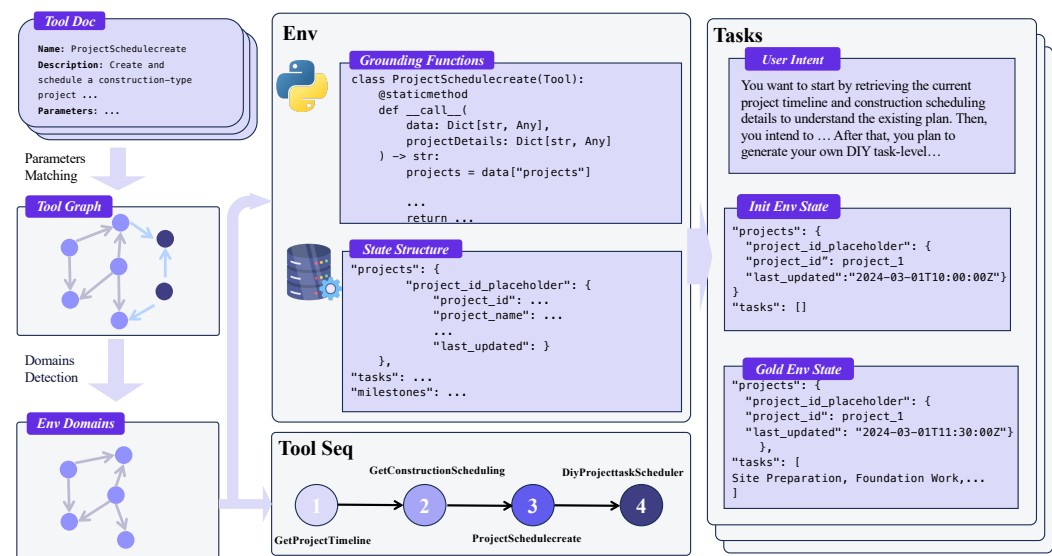

Figure 1: The overview of the environment automatic build, and agentic environment construction. Step 1, a large set of raw tool schemata are matched according to the vector similarity of their parameters, thereby constructing a tool graph; Step 2, use a community partitioning algorithm to divide the set of tools within a domain, then perform random walks to obtain tool sequences; Step 3, construct executable Python functions and a state repository from the tools.

## 2.1 ENVIRONMENT AUTOMATIC BUILD

Building upon this design principle, we propose a systematic pipeline for leveraging a diverse set of tools as shown in Figure 1. We begin with **scenario collection**, which gathers a large corpus of real-world tools; proceed to **tool dependency graph modeling**, which induces well-structured domain partitions and distributions; and finally employ **function schema programmatic materialization**, which maps tool operations onto database interactions, thereby enabling the construction of the overall environment.

**Scenario Collection** We collected more than 30,000 APIs from ToolBench (Qin et al., 2023; Guo et al., 2024), API-Gen (Prabhakar et al., 2025b) and online tool repository. After applying rigorous filtering, including the removal of low-quality APIs and subsequent refinement, we rewrite some API descriptions to incorporate explicit input–output specifications (Fang et al., 2025). Building on this, we further constructed tool compositions by systematically exploiting the input–output relationships among APIs. This process ultimately resulted in API pools $\Theta_F$ whose size = $N$ (over 30,000), providing a reliable foundation for subsequent experiments and analysis.

**Tool Dependency Graph Modeling** We construct a tool graph in which nodes are tools and edges encode compositional compatibility induced by function parameters. A tool $func$ consists of a description $D_{func}$ and a list of parameters $P_{func}$. For a pair of tools, we can extract their respective parameter lists and convert them into vector representations $\phi$ to compute their cos-similarity. If the similarity exceeds a predefined threshold $\tau$, we consider there to be a dependency relationship between the two tools. Accordingly, we insert an edge $E$ between them in our graph.

$$E = \big\{ (i, j) \mid \text{sim}(\phi(P_{func_i}), \phi(P_{func_j})) > \tau, \ i \neq j \big\} \tag{1}$$

Domain partitioning then reduces to a graph clustering problem. We employ Louvain community detection (Blondel et al., 2008) to identify coherent tool communities that serve as domains. For a segmented tool set, since parameter matching relies solely on vectorization and considers only individual parameter information, the overall inter-tool dependencies may be difficult to capture. Therefore, for tools within a given domain, we further employ an LLM to systematically examine the

dependencies between each pair of tools, thereby further improving the accuracy of edges in the tool graph. In total, we obtained $M$ domains (exceeding 1,000).

**Function Schema Programmatic Materialization** We first leverage the parameters of all tools within a domain to generate a domain-specific database structure, which serves as the underlying state for subsequent tool operations. After obtaining the domain-specific tool set and the corresponding database schema in the previous stage, we can formalize each tool in python code, enabling it to perform `read-write` operations over the database schema. Interestingly, when generating database structures and formalizing code within specific domains of $\tau$-bench, we observe through manual inspection that our outputs exhibit a high degree of consistency with the official implementations provided by $\tau$-bench (Yao et al., 2024).

## 2.2 Agentic Task Construction

We construct trajectories via forward simulated agent–human interplay, which allows us to fully simulate the environment, the user, and the agent. The critical step is to synthesize agentic tasks that elicit human tool usage while ensuring that the resulting trajectories remain **verifiable**. Concretely, we first initialize an environment state based on the domain-specific database schema, while encouraging as much diversity as possible in the initial state. Next, we sample logically coherent tool sequences from the domain's tool graph, specifically by constructing a directed dependency graph over APIs and traversing it to obtain valid sequences. Starting from a randomly selected initial node, we conduct a directed walk until either the maximum execution steps are reached or a node with no outgoing edges is encountered. This process yields a logically coherent tool sequence. For each step, we generate the corresponding arguments and perform the actual tool call, grounding the operations directly on the database and continuously tracking the evolving database state. This procedure enables verifiability at two complementary granularities: *(i)* database-level state consistency and *(ii)* exact matching of tool sequences.

# 3 Agent Experience Learning

We leverage user intent to drive interactions that yield agent experiences, and train the model through a two-phase process.

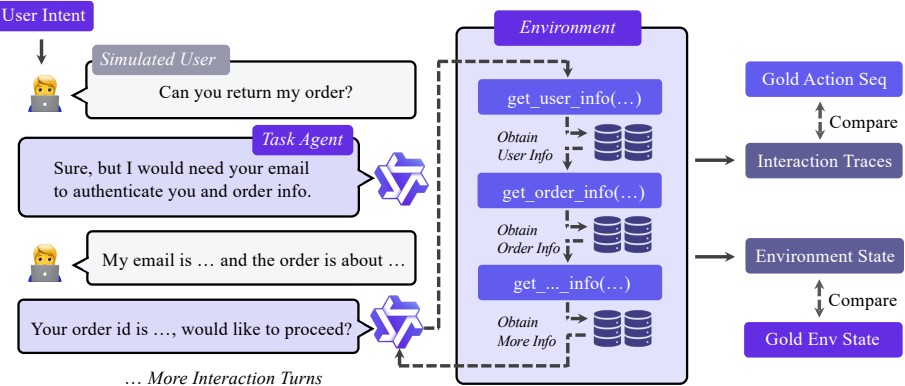

Figure 2: The agent interacts with the simulated user and changes the environment state through the generated functions.

## 3.1 Human–Agent Interplay for Experience Collection

**Interplay** Motivated by Yao et al. (2024), once we have constructed an agentic task, we proceed to perform human-agent interplay in the environment. Specifically, we instantiate a simulated user tasked with fulfilling a given overall intent. The agent then leverages domain-specific tools to address the user's needs, continuing the interaction until the simulated user deems the task complete. This

setup enables us to conduct **end-to-end simulation**, encompassing user simulation, agent, and environment, yielding a highly scalable framework. Each completed interaction trace constitutes an agent experience, which can subsequently be used for training. Importantly, since we possess both the gold tool sequences and arguments for the overall intent and the final environment state, we can apply these as supervision signals for experience filtering. All simulated trajectories are generated using the open-source model Qwen3-235B-A22B-Thinking.

**Filtering**  We adopt a three-stage funnel-based trajectory filtering framework consisting of *validity control*, *environment state alignment*, and *function calling exact match*.

- *Validity control*, removes invalid interaction trajectories to ensure well-formed alternating user assistant exchanges. Additionally, we apply an $n$-gram-based filtering procedure to eliminate severely repetitive reasoning segments. In such cases, we discard these data points.
- *Environment state alignment* retains only those trajectories whose final database state matches the golden state after the interplay, thereby validating the effectiveness of write operations. The filtering granularity at this stage is the **database/environment level**.
- *Function calling exact match* serves as the most stringent filtering stage, where the granularity is the **tool sequence**. Since a tool sequence consisting entirely of read operations without any write operations would cause state-based filtering to fail, we adopt a stricter exact match approach for filtering in such cases. A trajectory is preserved only if the sequence of invoked tools and arguments exactly matches the overall intent, ensuring high-fidelity supervision.

It is worth noting that we do not filter out trajectories in which tool calls return errors. Thanks to the aforementioned filtering framework, such trajectories may still accomplish the intended goal despite intermediate failures. Retaining them in the training data helps improve the robustness of the model.

## 3.2 AGENTIC EXPERIENCE LEARNING

**Agentic Fine-tuning**  Given agent-human interplay experience trajectory $\mathcal{H} = (h_0, a_1, \ldots, a_{n-1}, h_n, a_0)$, where each human instruction is denoted by $h_t$ at $t$-round interaction, and each assistant turn $a_t$ is decomposed as $a_t = (\tau_t, \rho_t, y_t)$. Here, $\tau_t$ represents the function call tokens, $\rho_t$ the tool response tokens, and $y_t$ the assistant response tokens. Our training objective is to optimize only the tool calls and assistant responses, while human instructions $h_i$ and tool responses $\rho_t$ are excluded from the loss. Formally, given an autoregressive model $p_\theta(x_k \mid x_{<k})$, we define the loss as

$$\mathcal{L}(\theta) = -\frac{1}{\sum_{k=1}^{|\mathcal{H}|} \mathbb{I}[k_i \in \mathcal{T}]} \sum_{k=1}^{|\mathcal{H}|} \mathbb{I}[x_k \in \mathcal{T}] \cdot \log \pi_\theta\left(x_i \mid x_{<k}\right), \tag{2}$$

where $x_k$ denotes the $k$-th token in the trajectory, $\pi_\theta$ is the model distribution, $\mathbb{I}[\cdot]$ is the indicator function, $\mathcal{T}$ is the set of tokens belonging to tool calls $\tau$ or assistant responses $y$. In practice, all tokens in $\rho_i$ and $h_i$ are masked out from supervision but remain visible in the context $x_{<k}$. This ensures that the model conditions on tool responses and human instructions, while gradients are only propagated through assistant-generated tool calls and natural-language responses.

**Two-stage Experience Learning**  In the first phase, the agent is trained to acquire fundamental skills for tool usage and user interaction. We focus on general domains where a broad set of tools and tasks are available, allowing the agent to develop a robust understanding of when and how to invoke function calls, as well as how to integrate tool outputs into coherent user-facing responses. This stage emphasizes breadth and generality, ensuring that the agent builds a versatile foundation of agentic behaviors before domain-specific specialization. In the second phase, the agent undergoes fine-grained training in vertical domains, where tasks, tools, and user intents exhibit domain-specific characteristics. In our setting, this stage primarily focuses on the $\tau$-Bench and $\tau^2$-Bench. By grounding the learning process in realistic scenarios within a target domain, the agent refines its ability to select tools, parameterize calls, and produce responses that are accurate, contextually appropriate, and aligned with domain-specific goals. This specialization ensures a smoother adaptation of agentic capabilities, enabling the agent to operate effectively in real-world, task-oriented environments.

## 4 EXPERIMENTS

### 4.1 SETUP

**Benchmarks** We evaluate our methods on three established agentic benchmarks: $\tau$-bench, $\tau^2$-Bench, and ACEBench-en. For $\tau$-Bench (covering the `retail` and `airline` domains) and $\tau^2$-Bench (spanning the `retail`, `airline`, and `telecom` domains), we adopt the $\text{pass}^1$ metric for evaluation and additionally analyze the trend of $\text{pass}^k$, following the protocols in Yao et al. (2024); Barres et al. (2025).

For ACEBench-en, we report results across the `Normal`, `Special`, and `Agent` categories, as well as the `Overall` performance, using the accuracy metric.

**Baselines** We compare our trained series models against the following types: *closed-sourced large language model*, including Gemini-2.5-pro (Comanici et al., 2025), Claude-Sonnet-4 (Anthropic, 2025), GPT-o3, GPT-o4-mini (OpenAI, 2025b), and GPT-5 (with thinking) (OpenAI, 2025a); *open-sourced large language model*: GPT-OSS-120B-A5B (Agarwal et al., 2025), Deepseek-V3.1-671B-A37B (DeepSeek-AI, 2024), Kimi-K2-1T-A32B (Team et al., 2025), Qwen3-Thinking-235B-A22B (Team, 2025b), Seed-OSS-36B (Team, 2025a), Qwen-Coder-30B-A3B (Hui et al., 2024), and xLAM-2 model series (Prabhakar et al., 2025a).

**Backbones** We train the AgentScaler model series by training on Qwen3 models (Team, 2025b) of varying scales. Specifically, AgentScaler-4B and AgentScaler-30B-A3B are trained on Qwen3-Thinking-4B-2507 and Qwen3-Thinking-30B-A3B-2507, respectively, while AgentScaler-8B is trained on Qwen3-8B.

### 4.2 EXPERIMENTAL RESULTS

Table 1: **Main results** on $\tau$-Bench, $\tau^2$-Bench, and ACEBench-en.

| Model | $\tau$-bench | | $\tau^2$-Bench | | | ACEBench-en | | | |
|---|---|---|---|---|---|---|---|---|---|
| | Retail | Airline | Retail | Airline | Telecom | Normal | Special | Agent | Overall |
| *Closed-Source Large Language Models* | | | | | | | | | |
| Gemini-2.5-pro | 68.7 | 44.0 | 67.5 | 56.0 | 27.2 | 76.7 | 90.0 | 63.4 | 78.2 |
| Claude-Sonnet-4 | 73.9 | 40.0 | 67.5 | 54.0 | 47.4 | 79.9 | 87.3 | 42.5 | 76.1 |
| GPT-o3 | 70.4 | 52.0 | 80.2 | 64.8 | 58.2 | 78.3 | 86.7 | 63.3 | 78.2 |
| GPT-o4-mini | 70.4 | 46.0 | 70.2 | 56.0 | 46.5 | 79.9 | 84.0 | 60.0 | 77.9 |
| GPT-5-think | 78.3 | 44.0 | 81.1 | 62.6 | 96.7 | 76.7 | 85.3 | 32.5 | 72.2 |
| *Open-Source Large Language Models* | | | | | | | | | |
| GPT-OSS-120B-A5B | 67.8 | 49.2 | 57.0 | 38.0 | 45.6 | 79.1 | 84.0 | 50.8 | 76.0 |
| Deepseek-V3.1-671B-A37B | 66.1 | 40.0 | 64.9 | 46.0 | 38.5 | 80.3 | 62.0 | 40.8 | 69.3 |
| Kimi-K2-1T-A32B | 73.9 | 51.2 | 70.6 | 56.5 | 65.8 | 78.9 | 81.3 | 65.0 | 77.4 |
| Qwen3-Thinking-235B-A22B | 67.8 | 46.0 | 71.9 | 58.0 | 45.6 | 72.1 | 84.0 | 39.1 | 70.2 |
| Seed-OSS-36B | 70.4 | 46.0 | 68.4 | 52.0 | 41.2 | 79.1 | 82.0 | 58.4 | 76.7 |
| Qwen-Coder-30B-A3B | 68.7 | 48.0 | 60.5 | 42.0 | 30.7 | 74.0 | 41.3 | 24.1 | 57.5 |
| xLAM-2-8B-fc-r | 58.2 | 35.2 | 55.3 | 48.0 | 11.4 | 58.8 | 0.0 | 5.0 | 34.8 |
| xLAM-2-32B-fc-r | 64.3 | 45.0 | 55.3 | 52.0 | 16.7 | 69.2 | 24.7 | 13.4 | 52.5 |
| xLAM-2-70B-fc-r | 67.1 | 45.2 | 61.4 | 56.0 | 14.0 | 57.1 | 5.3 | 38.4 | 36.5 |
| Qwen3-Thinking-4B | 59.1 | 52.5 | 56.1 | 52.0 | 28.7 | 43.3 | 84.7 | 11.7 | 49.5 |
| Qwen3-8B | 45.2 | 25.0 | 41.2 | 30.5 | 23.5 | 71.4 | 75.3 | 29.1 | 65.9 |
| Qwen3-14B | 45.7 | 31.0 | 48.0 | 30.0 | 26.9 | 66.9 | 84.0 | 44.2 | 68.0 |
| Qwen3-Thinking-30B-A3B | 67.8 | 48.0 | 58.8 | 58.0 | 26.3 | 64.7 | 86.7 | 42.8 | 67.2 |
| **AgentScaler-4B** | 64.3 | 54.0 | 62.3 | 56.0 | 48.2 | 70.3 | 76.7 | 30.8 | 65.9 |
| **AgentScaler-8B** | 50.4 | 42.0 | 58.8 | 44.0 | 45.4 | 69.2 | 76.7 | 44.2 | 67.4 |
| **AgentScaler-30B-A3B** | 70.4 | 54.0 | 70.2 | 60.0 | 55.3 | 76.7 | 82.7 | 60.0 | 75.7 |

**Main Results** From Table 1, we observe that closed-source large language models (LLMs) still maintain a clear performance advantage, consistently achieving the highest scores across most domains and benchmarks. This demonstrates the strength of industrial-scale training pipelines and proprietary optimization strategies. Nevertheless, our proposed AgentScaler achieves a remarkable level of performance given its lightweight parameter scale. Specifically, it surpasses most open-source baselines with fewer than 1T parameters, establishing a new state-of-the-art across $\tau$-bench, $\tau^2$-Bench, and ACEBench-en.

Notably, **AgentScaler-4B** achieves performance on par with 30B-parameter models despite using the fewest parameters, highlighting the agentic potential of compact LLMs. Moreover, **AgentScaler-30B-A3B** delivers results that are comparable to trillion-parameter open-source models and, in several domains, approach those of closed-source counterparts. These findings highlight the efficiency of our approach: agentic capabilities can be effectively learned and deployed even in relatively compact models, enabling competitive performance

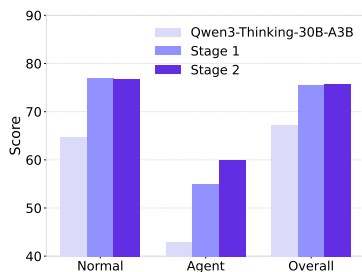

Figure 3: Performance comparison on the `Normal`, `Agent`, and `Overall` subsets of ACEBench-en for two-stage training models.

without relying on massive parameter counts. This advantage makes AgentScaler particularly well-suited for practical deployment in resource-constrained or latency-sensitive scenarios.

**Ablation Study**   We further conduct an ablation analysis to examine the effect of the proposed two-stage agent experience learning framework on ACEBench-en. As shown in Figure 3, both Stage 1 and Stage 2 training substantially improve performance over the base model (Qwen3-Thinking-30B-A3B) across all subsets. And through multi-steps agent training in Stage 2, the model's score on the agent set has further improved, and the overall score has also increased. These results validate the design of the two-phase training pipeline: general foundation learning is critical for establishing tool-usage competence, and subsequent domain-specialization further consolidates and contextualizes these capabilities.

## 5 ANALYSIS

**Our synthetic data approach enables efficient knowledge transfer and strong robustness and generalization.** We further evaluate our models on ACEBench-zh, which represents an out-of-distribution (OOD) scenario relative to the training setup. The observed drops, AgentScaler-4B on special, AgentScaler-8B on normal, and AgentScaler-30B-A3B on special, are likely attributable to these OOD effects. As shown in Ta-

Table 2: The **results** on ACEBench-zh.

| Model | ACEBench-zh | | | |
|---|---|---|---|---|
| | Normal | Special | Agent | Overall |
| Qwen3-Thinking-4B | 34.7 | 85.3 | 6.7 | 43.9 |
| AgentScaler-4B | 70.8+36.1 | 70.0-15.3 | 38.4+31.7 | 65.6+21.7 |
| Qwen3-8B | 80.3 | 72.7 | 35.0 | 71.3 |
| AgentScaler-8B | 75.2-5.1 | 79.3+6.6 | 58.4+23.4 | 73.7+2.4 |
| Qwen3-Thinking-30B-A3B | 73.4 | 86.7 | 55.8 | 74.2 |
| AgentScaler-30B-A3B | 85.3+11.9 | 83.3-3.4 | 64.1+8.3 | 81.5+7.3 |

ble 2, the AgentScaler models consistently outperform their Qwen baselines across all scales in terms of overall score. In particular, AgentScaler-30B-A3B achieves the best overall score of 81.5, demonstrating strong improvements in both the Normal and Agent subsets, while maintaining competitive performance on the Special subset. Notably, the small Qwen3-4B model demonstrated a remarkable improvement in agentic capabilities after the two-stage training, with its score surging from 6.7 to 38.4 and substantial gain of 21.7 points in the overall score. This offers valuable insights into effectively training compact models for complex function calling tasks in real-world applications. Our evaluation setup does include domain- and format-level OOD generalization, not only cross-lingual robustness. First, ACEBench-en can be also seen an OOD evaluation for our system. The environments we construct use APIs sourced from ToolBench and API-Gen, which are not overlapping with the tool domains or schema structures in ACEBench-en. Therefore, ACEBench-en evaluates the model's ability to generalize to unseen tool domains, rather than only testing in-domain performance. Second, the tool-calling format itself is out-of-distribution. Our training is performed in the Qwen3-Hermes tool-calling format, while ACEBench adopts its own custom parser format. Achieving strong performance under the ACEBench-en parsing and tool-calling rules demonstrates format-level generalization, showing that the model is not overfitted to a single schema or interaction protocol. Third, ACEBench-zh provides an additional cross-lingual generalization test, further validating robustness across languages.

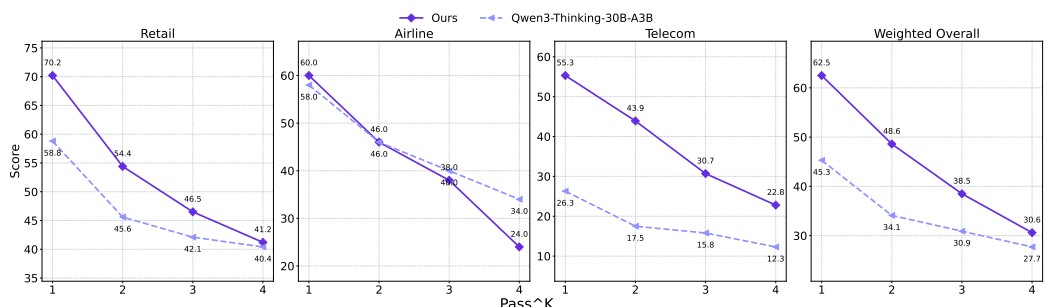

Figure 4: Pass^k metric results across all domains in the $\tau^2$-Bench.

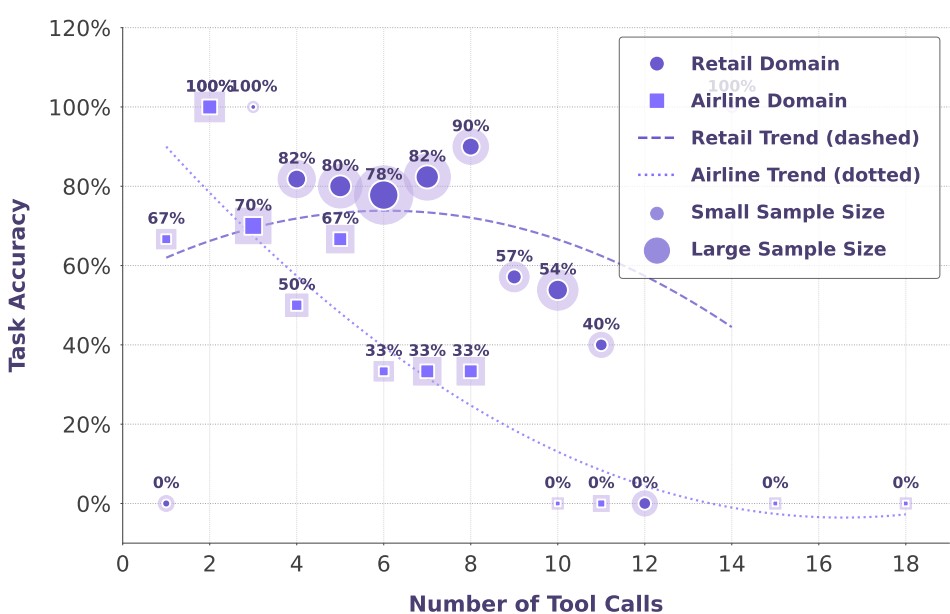

Figure 5: Accuracy by tool call count on $\tau$-bench.

**AgentScaler shows the strong consistency, stability.** To assess the stability of AgentScaler, Figure 4 reports the pass^k metric on the $\tau^2$-Bench, which denotes the accuracy achieved when the model correctly answers the same question in all k independent trials. According to the experimental results, the weighted overall score of AgentScaler-30B-A3B consistently surpasses that of Qwen3-Thinking-30B-A3B across all evaluated pass^k settings, indicating a substantial performance advantage of our model over Qwen3-Thinking-30B-A3B. Moreover, a clear downward trend in scores is observed as k increases, suggesting that the stability of existing LLMs remains a considerable challenge.

**Long-horizon tool calling remains a fundamental challenge for agentic models.** To further analyze the model's long-horizon tool-calling capability, we constructed a scatter plot on the $\tau$-bench dataset showing the relationship between the number of tool calls in each trajectory and the corresponding trajectory accuracy, with a dashed line indicating the trend. As illustrated in Figure 5, there exists a clear negative correlation between the number of tool calls and task accuracy. Our AgentScaler models exhibit this trend as well, underscoring that handling extended tool-use chains is still an open problem that we plan to address in future work.

**Scaling Law of Environments.** We conduct a preliminary verification of scaling laws during the continued pre-training stage. Specifically, when incorporating our dataset into the continued pre-training phase, the Qwen3-30B-A3B-Thinking model achieved a 2.8-point improvement on ACEBench-en.

## 6 RELATED WORK

### 6.1 TOOL-USE ENVIRONMENTS

The construction of tool-use environments primarily involves three approaches: real-world environments, LLM–simulated environment, and Simulated Environments based on a state config. Using real-world environments (Qin et al., 2023; Song et al., 2023; Mastouri et al., 2025; Wu et al., 2025b) to invoke actual tools yields the most authentic feedback and enhances the model's robustness in practical applications. However, this requires frequent calls to MCP services, resulting in high costs and significant time overhead. Moreover, maintaining a highly available and stable MCP service is often difficult, posing major challenges for agentic data generation and online RL training of models.Many works use LLM-generated responses to simulate environments as a source of tool responses (Qin et al., 2024; Lu et al., 2024; Sun et al., 2025). By leveraging strong or fine-tuned LLMs, these approaches generate plausible responses given a tool call. However, such methods struggle with issues like hallucination and inconsistent response variability. To address the limitations of the above two approaches, some recent work (Ye et al., 2025b; Yao et al., 2024; Barres et al., 2025; Prabhakar et al., 2025b; Ye et al., 2025a) proposes building an offline tool execution environment for LLM training and evaluation. On one hand, offline environments avoid calling real tools, significantly reducing response generation cost and latency. On the other hand, mocked tool usage in such environments can still interact with real databases or state files through actual execution. However, these methods are more commonly applied in LLM evaluation rather than training, as constructing a reliable tool suite and a high-fidelity execution environment typically requires substantial manual effort. Furthermore, it is difficult to automatically validate the quality of such environments without human involvement, making scalability a significant challenge. Our approach enables domain scalability through sampling from a toolgraph, and eliminates the need for human intervention via a rigorous, rule-based validation pipeline. This makes scalable construction of tool execution environments feasible.

### 6.2 TOOL LEARNING

To enhance the agentic capabilities and tool-calling abilities of models, many works have attempted to improve tool utilization through various approaches. For instance, xLAMs (Prabhakar et al., 2025b; Zhang et al., 2024) and ToolAce (Liu et al., 2024a) leverage large-scale agentic data synthesis pipelines to generate high-quality training data and thereby boost model performance. DiaTool-DPO (Jung et al., 2025) employs DPO to enable models to learn from multi-turn positive and negative trajectories. Meanwhile, Tool-RL (Qian et al., 2025), Tool-N1 (Zhang et al., 2025) utilize reinforcement learning (RL) algorithms to enhance both the tool-calling proficiency and generalization ability of models, further pushing the performance boundaries beyond supervised fine-tuning. Overall, whether relying on agentic data synthesis or online interaction with environments via RL training, a stable, reliable, and scalable execution environment is essential. For example, Kimi-K2 (Team et al., 2025) uses a tool simulator during Agentic Data Synthesis to obtain observations for multi-turn trajectories. Our method not only leverages accurately simulated tool environments to collect trajectories but also introduces verifiable environmental state changes, making each simulation response more reliable. Furthermore, we propose a state change based environment validation strategy, enabling a robust filtering mechanism for large-scale agentic data synthesis.

## 7 CONCLUSION

In this work, we presented a principled pipeline for advancing general agentic intelligence through systematic environment scaling and agent experience learning. By programmatically materializing tools as executable code and grounding them in database-structured environments, our approach enables large-scale construction of verifiable trajectories. Building on these environments, we introduced a two-stage agent experience learning framework that first equips agents with fundamental tool-usage capabilities and then specializes them for domain-specific contexts. Extensive experiments on three representative benchmarks, $\tau$-bench, $\tau^2$-Bench, and ACEBench, demonstrate the effectiveness of our pipeline. Notably, our AgentScaler family achieves state-of-the-art performance among open-source models under 1T parameters, and in several cases reaches parity with much larger or closed-source counterparts.

Looking ahead, we believe our work highlights the importance of scalable environment construction and verifiable agentic experience for fostering robust and generalizable language agents. Future directions include integrating reinforcement learning on top of our fully simulated environments and extending our pipeline to broader modalities and real-world deployment scenarios.

## LIMITATION

Although our proposed framework has demonstrated promising results, several limitations remain, which point to ongoing efforts and potential directions for future work.

**Reinforcement-Learning Integration**  Although the current system relies solely on two-stage supervised fine-tuning, the simulator we have built offers deterministic, low-latency feedback that is ideal for reinforcement-learning optimization. In future iterations we plan to add an RL stage using policy gradient methods, to refine the agent's long-horizon decision-making and further improve its emergent, agentic capabilities.

**Model Scale**  Another limitation of our current work lies in the model scale. Our method has so far only been validated on a 30B-scale architecture, without extension to larger models exceeding 200B or even trillion-parameter scales. While prior work (Belcak et al., 2025) emphasizes that "small language models are the future of agentic AI," we share the view that training agentic capabilities in relatively smaller models is particularly meaningful. Such models are easier to deploy on edge devices, enable broader applicability across diverse scenarios, and offer faster response times.

## ETHICS STATEMENT

This study strictly adheres to established ethical guidelines at every stage. During the tool-collection phase, all code, models, and utilities were obtained exclusively from publicly available, open-source repositories or officially documented APIs released under permissive licenses. No proprietary or restricted software was employed. Furthermore, every data point used in the experiments was synthetically generated through algorithmic means. Crucially, no personally identifiable information was collected, accessed, or produced at any time.

## REPRODUCIBILITY STATEMENT

To ensure reproducibility, we provide a thorough description of data construction, training procedures, and evaluation details. Section 2 presents a step-by-step account of how we automatically build the agent environment with an LLM. Section 3 exhaustively covers the collection and filtering of trajectory data, and elaborates on the two-stage training pipeline that yields our final agent model. The evaluation setting and fine-grained metrics can be found in Appendix B.

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

## A THE USE OF LARGE LANGUAGE MODEL

We affirm that Large Language Models are employed solely as an assisted tool to refine wording and sentence structure during our paper writing process. Their use in the experiments is strictly for scientific research purposes, and all such usage has been explicitly documented in our Experimental Settings and Reproducibility Statement. No other reliance on LLMs is involved in this work.

## B  EXPERIMENT SETTINGS

During the agentic experience learning stage, all models in the AgentScaler series — including 4B, 8B, and 30B-A3B — were trained for three epochs with a batch size of 128. The model context length was set to 32768. Subsequently, we directly used the model from the final checkpoint. Additionally, the learning rate was set to 7e-6, and a warm-up strategy was employed.

During the evaluation stage, we strictly followed all the official guidelines of the benchmarks. All baseline models were assessed using the benchmarks' default configurations. For the AgentScaler series of models, we set the temperature parameter to 0.6, the top-p value to 0.95, and the top-k value to 20. Moreover, to speed up model inference, we deployed the model using vLLM (Kwon et al., 2023).

