# OpenReview forum: "Towards General Agentic Intelligence via Environment Scaling"
_ICLR.cc/2026/Conference — ICLR 2026 Conference Withdrawn Submission_

### Official Review · Reviewer_rX7d · 2025-10-30

**Soundness:** 2
**Presentation:** 3
**Contribution:** 2
**Rating:** 4
**Confidence:** 4

**Summary:**

This paper introduces AgentScaler, a framework for advancing general agentic intelligence through environment scaling and teacher-free experience learning. The authors propose an automatic pipeline that constructs diverse tool-use environments by clustering 30k+ APIs into functional domains, materializing them as executable code with mock database schemas, and generating synthetic agent–human trajectories through simulated interactions. Agents are then fine-tuned in two stages: a general foundation phase and a domain-specific specialization phase.
Empirical evaluations on τ-Bench, τ²-Bench, and ACEBench show consistent improvements over Qwen3 baselines, with further cross-lingual tests on ACEBench-zh. The work positions itself as a scalable, verifiable alternative to teacher-model distillation.

**Strengths:**

- The environment construction pipeline is genuinely novel. It derives “gold” supervision signals directly from environment logic (database state transitions and tool dependency graphs) rather than relying on stronger teacher models or human annotation. Allowing it to scale nicely

- Representing tools as read/write operations over structured domains is an elegant unifying abstraction that allows programmatic  environment generation.

- The three-stage trajectory filtering process (validity → state alignment → exact match) provides a credible mechanism for ensuring quality and consistency in synthetic data.

**Weaknesses:**

- The paper never explicitly states which model generated the simulated trajectories. It appears that the base model under training were used for generation, but this should be made explicit to rule out hidden distillation.

- Limited notion of “generalization”. The only OOD test (ACEBench-zh) evaluates cross-lingual robustness, not transfer to unseen tool domains or schema structures. Because τ-Bench–derived tools appear in both training and evaluation, true domain generalization remains untested.

- While overall metrics improve, per-domain results fluctuate substantially. For example, in ACEBench-zh the “Special” subset consistently drops across model sizes, and in τ²-Bench some domains improve dramatically while others regress. No analysis is offered to explain these discrepancies.

- The pass^k curves show higher pass^1 accuracy but a faster decline as k increases, ultimately converging with the baseline. This indicates better one-shot precision, not improved stability or consistency across runs.

- The paper offers no insight into why certain domains or subsets improve more than others. Missing are per-domain data distribution analyses, qualitative error cases, or ablations on environment size or filtering stages.

- The environment builder reproduces τ-Bench-like schemas (“high consistency with official implementations”), suggesting potential overlap between training and benchmark environments, which weakens the validity of reported gains.

**Questions:**

- Which model was used as the agent during synthetic experience generation — the same Qwen3 base model or a stronger teacher? It would be helpful clarify explicitly.

- Data–evaluation overlap: How do you ensure that APIs, schemas, or tool sequences from τ-Bench or τ²-Bench are not directly included in the training environments? Do you plan to evaluate on unseen tool domains or unseen function schemas to substantiate claims of “general agentic intelligence”?

---

> ### Author Response · Authors · 2025-11-21
>
> > **Weakness1 & Question1: Model for Trajectories**
>
> Thank you for raising this point. To clarify, all simulated trajectories were generated using the open-source model **Qwen3-235B-A22B-Thinking**. Furthermore, as shown in Figure 1, the final AgentScaler models **consistently surpass Qwen3-Thinking-235B-A22B across all evaluated benchmarks**.
> This provides strong evidence that our synthesized data is **not merely replicating or distilling the behavior of the trajectory generator**.
> Instead, it enables the smaller models to generalize beyond the capabilities of the teacher model used for data synthesis.
>
> > **Weakness2 & Question2: Generalization**
>
> We would like to clarify that our evaluation setup does include domain- and format-level OOD generalization, not only cross-lingual robustness.
>
> - First, ACEBench-en can be also seen **an OOD evaluation** for our system.
> The environments we construct use APIs sourced from ToolBench and API-Gen, which are not overlapping with the tool domains or schema structures in ACEBench-en.
> Therefore, ACEBench-en evaluates the model’s ability to generalize to unseen tool domains, rather than only testing in-domain performance.
>
> - Second, the **tool-calling format itself is out-of-distribution**.
> Our training is performed in the Qwen3-Hermes tool-calling format, while ACEBench adopts its own custom parser format. Achieving strong performance under the ACEBench-en parsing and tool-calling rules demonstrates format-level generalization, showing that the model is not overfitted to a single schema or interaction protocol.
>
> - Third, ACEBench-zh provides an additional **cross-lingual generalization test**, further validating robustness across languages.
>
> > **Weakness3 & Weakness5: Performance Drop on ACEBench-zh**
>
> Thank you for pointing out the performance fluctuations in Table 2.
> We would like to clarify that Table 2 reflects an out-of-distribution (OOD) evaluation, where we report results on ACEBench-zh, which differs in distribution from the training data of the base models.
> The observed drops, AgentScaler-4B on `special`, AgentScaler-8B on `normal`, and AgentScaler-30B-A3B on `special`, are likely **attributable to these OOD effects**.
> Importantly, the `agent` subset exhibits substantial improvements, and the overall performance also increases across model sizes.
> These trends suggest that, despite subset-specific OOD variability, AgentScaler consistently enhances the models’ function-calling ability.
> We will include additional analysis in the revised version to clarify the evaluation and to better interpret the subset-level differences.
>
> > **Weakness4: Pass^k  Metric Performance**
>
> Thank you for pointing this out. We agree that the stability of pass^k as k increases remains a challenging and largely unresolved issue. A clear downward trend emerges with larger k, indicating that the stability of current LLMs is still a significant limitation, an observation also supported by several prior studies[1,2].
> That said, our method still **consistently outperforms the baseline across different k**, although the gains naturally diminish as
> k grows. We acknowledge that achieving substantial improvements at larger k is difficult, and we believe continued efforts from the community will be required to make further progress on this front.
>
> [1]. APIGen-MT: Agentic PIpeline for Multi-Turn Data Generation via Simulated Agent-Human Interplay
>
> [2].  $\tau^2$-Bench: Evaluating Conversational Agents in a Dual-Control Environment
>
> > **Weakness5: Quantitative Analysis of Environment Scaling and Case Study**
>
> We conduct a preliminary verification of scaling laws during the continued pre-training stage. Specifically, when incorporating our dataset into the continued pre-training phase, the Qwen3-30B-A3B-Thinking model achieved a 2.8-point improvement on ACEBench-en.
>
> As for scaling laws in the post-training stage, due to time constraints we are unable to complete the full set of experiments before this submission. We plan to finish these studies and include the results in a future revision.

---

> ### Author Response · Authors · 2025-11-21
>
> > **Weakness6 & Weakness2: Data–evaluation Overlap**
>
> Thank you for raising this important concern regarding potential data–evaluation overlap. We address it from several angles:
>
> - First, our use of the environment builder to reproduce $\tau$-Bench–like schemas is intended **purely as a probe experiment to demonstrate that the builder can generate coherent and reasonable environments**. This diagnostic step does not imply that the generated environments are identical to the official benchmarks.
>
> - Second, we take **explicit measures** to ensure that APIs, schemas from $\tau$-Bench or $\tau^2$-Bench are not reproduced in the training set. This includes both automated similarity checks and human verification to rule out accidental overlap.
>
> - Third, even in the hypothetical case where certain API names coincide, it remains **extremely unlikely** that identical tool sequences would be generated. Our environment synthesis incorporates substantial randomness, and more importantly, **the underlying databases and parameter spaces of the generated environments differ from those** in the benchmarks. Consequently, identical API calls would still yield **different tool arguments and execution traces**.
>
> For these reasons, we are confident that **no meaningful data–evaluation leakage occurs**.
>
> **Thank you for your careful feedback. We will make sure to address any further questions with the same level of rigor and attention.**

---

### Official Review · Reviewer_HUX4 · 2025-10-31

**Soundness:** 3
**Presentation:** 3
**Contribution:** 3
**Rating:** 6
**Confidence:** 4

**Summary:**

The paper introduces AgentScaler, a pipeline for developing general agentic intelligence through large-scale simulated tool-use environments and two-stage training (general → domain-specialized). It automatically constructs verified tool-use trajectories from 30K+ APIs, clusters them into domains, and trains models (4B–30B) that outperform comparable open LLMs on τ-Bench, τ²-Bench, and ACEBench.

**Strengths:**

- The paper presents a novel and automated approach to transform large-scale APIs into executable and verifiable environments, significantly improving reproducibility and coverage for agentic training.
  - The two-stage training strategy, which separates general capability building from domain specialization, is well-motivated and empirically validated to enhance performance across benchmarks.
  - The results demonstrate strong effectiveness, with compact models (e.g., 30B) achieving performance close to proprietary systems, showing that the proposed pipeline is efficient and scalable.

**Weaknesses:**

- The methodological novelty of the paper is relatively limited. The two-stage training strategy (general-to-domain specialization) closely resembles existing agent fine-tuning pipelines such as AgentFlan, ToolAce
  - The paper does not compare its environment construction approach or data generation pipeline with other existing frameworks, nor does it analyze in detail how environment scaling quantitatively affects model performance.

**Questions:**

- Could the authors provide a more detailed quantitative analysis of how environment scaling influences model performance? For example, how does expanding the number or diversity of simulated environments affect the model’s tool-use accuracy, generalization, or stability across benchmarks?

---

> ### Author Response · Authors · 2025-11-21
>
> > **Weakness1: Methodological Novelty**
>
> We agree that the two-stage training paradigm (general-to-domain specialization) is widely adopted in recent agent fine-tuning pipelines.
> More importantly, the primary principle we aim to convey goes beyond the training paradigm itself.
> Our main contribution lies instead in the scalable construction of large-scale tool-use environments, which substantially differs from prior environment-building and data-generation frameworks.
>
> - **Scalable simulated environments.**
> Our method builds high-fidelity tool environments that allow large-scale trajectory collection while maintaining deterministic, reproducible behavior across simulations.
>
> - **Verifiable environmental state changes.**
> A key innovation is that we introduce explicit, verifiable state transitions, ensuring that each tool response corresponds to a consistent underlying environment state. This makes the simulation feedback more reliable than prior loosely constrained or heuristic tool simulators.
>
> - **State-change–based environment validation for large-scale data synthesis.**
> We propose a state-validation mechanism that filters ill-formed or inconsistent trajectories at scale. This provides a robust and principled way to ensure data quality—an aspect that existing systems do not emphasize.
>
> - **Conceptual contribution: tools as environment read/write operations.**
> The core motivation of our paper—and the conceptual message we aim to convey to the community, is that tool use can be fundamentally viewed as read/write operations over an underlying database (enviroment).
> Under this abstraction, our environment-construction framework naturally scales to a wide range of tools and domains, enabling more systematic and interpretable agent learning.
>
> > **Qustion1: Quantitative Analysis of Environment Scaling**
>
> We conduct a preliminary verification of scaling laws during the continued pre-training stage. Specifically, when incorporating our dataset into the continued pre-training phase, the Qwen3-30B-A3B-Thinking model achieved a **2.8-point improvement on ACEBench-en**.
>
> As for scaling laws in the post-training stage, due to time constraints we are unable to complete the full set of experiments before this submission. We plan to finish these studies and include the results in a future revision.
>
> **We sincerely appreciate your thoughtful suggestions. We will make sure to address any additional issues that may arise.**

---

### Official Review · Reviewer_t5Y1 · 2025-11-01

**Soundness:** 2
**Presentation:** 3
**Contribution:** 2
**Rating:** 2
**Confidence:** 4

**Summary:**

In this paper, the authors argue that agents need to interact with diverse and scale environments to gain real world function calling abilities. Therefore, the authors build a scaled up environment framework to automatically construct simulated heterogeneous environments for agents to interact with. Then they finetuned agents on trajectories collected from their simulated environments and then evaluated on benchmarks.

**Strengths:**

- The authors provided a useful tool of scaling up simulated environments for agent to interact with.
- The authors performed comprehensive experiments by training on models of different sizes and evaluated on multiple benchmarks.
- The authors compared their models with various baselines.

**Weaknesses:**

- Seed-OSS-36B (which has similar model size with AgentScaler-30B-A3B) achieves higher performance on ACEBench-en (both normal and overall). Yet the authors still claimed that they achieved state of the art performance on all three benchmarks they evaluated on.
- In Table 2, it appears that after training, the performance of AgentScaler-4B dropped 15.3 on special, AgentScaler-8B dropped 5.1 on normal, AgentScaler-30B-A3B dropped 3.4 on special, yet the authors didn't explain why this might be happening. However, the authors still claimed that their trained model, AgentScaler, significantly enhances the models' function-calling capability.
- With the abundant existing environments that agents could interact with, the authors didn't make it clear why simulated environments are useful.

**Questions:**

- Why does performance dropped after training?

---

> ### Author Response · Authors · 2025-11-21
>
> > **Weakness1: Seed-OSS-36B Performance Comparison**
>
> Thank you for raising the comparison with Seed-OSS-36B. However, we must clarify that the statement in the review is based on a **factual misunderstanding** of the model sizes involved.
> Seed-OSS-36B is a dense 36B model, meaning 36B parameters are activated at inference time. In contrast, AgentScaler-30B-A3B is an MoE model that activates only 3B parameters during inference.
> Therefore, treating them as “similar model size” may lead to a **misunderstanding of their actual scale**.
>
> Regarding your comment on our “state-of-the-art” claim, we appreciate the observation.
> As you correctly pointed out, we do not surpass Seed-OSS-36B on ACEBench-en (both normal and overall), and thus it is not precise to state that we achieve state-of-the-art on all three evaluated benchmarks.
> Our intended claim is that we achieve state-of-the-art performance among models with comparable active parameter size, and outperform most models with total parameters below 1T.
> Achieving such performance with **only 30B parameters (3B active)** is, **in our view, a meaningful and non-trivial accomplishment**.
>
> We appreciate the reviewer for pointing this out and will **correct the phrasing** to avoid ambiguity.
>
> > **Weakness2 & Question1: Performance Drop on ACEBench-zh**
> Thank you for pointing out the performance fluctuations in Table 2.
> We would like to clarify that Table 2 reflects an out-of-distribution (OOD) evaluation, where we report results on ACEBench-zh, which differs in distribution from the training data.
>
> The observed drops, AgentScaler-4B on `special`, AgentScaler-8B on `normal`, and AgentScaler-30B-A3B on special, are likely **attributable to these OOD effects**.
>
> Importantly, the `agent` subset exhibits substantial improvements, and the overall performance also increases across model sizes.
> These trends suggest that, despite subset-specific OOD variability, AgentScaler consistently enhances the models’ function-calling ability.
>
> We will include additional analysis in the revised version to clarify the evaluation and to better interpret the subset-level differences.
>
> > **Weakness3: Motivation of Simulated Environments**
>
> We agree that there already exist many real-world environments for agent interaction[1].
> However, these real environments suffer from several practical limitations that make them unsuitable for large-scale and stable training.
>
> First, many real online APIs are unstable, **frequently affected by rate limits, fluctuating QPS capacity, and transient API failures**.
> Such instability prevents agents from receiving consistent tool feedback, which is crucial for effective learning.
> This limitation has also been highlighted by multiple recent works[2,3], motivating the development of simulated environments to enable reliable and large-scale agent training.
>
> Second, **simulated environments can provide deterministic, reproducible, and controllable tool feedback**, which real-world environments cannot guarantee.
> This stability is especially important for training models that rely on precise multi-step tool interactions[4,5,6].
>
> Finally, the core motivation of our paper, and the conceptual message we aim to convey to the community, is that **tool use can be viewed fundamentally as read/write operations over an underlying database (environment)**.
> Under this abstraction, building large-scale simulated environments becomes **not only feasible, but also an effective way to enable broad and scalable tool-use learning**.
>
> We will further clarify this motivation and the design principle of simulated environments in the revised version.
>
> [1]. ToolLLM: Facilitating Large Language Models to Master 16000+ Real-world APIs
>
> [2]. StableToolBench: Towards Stable Large-Scale Benchmarking on Tool Learning of Large Language Models
>
> [3]. StableToolBench-MirrorAPI: Modeling Tool Environments as Mirrors of 7,000+ Real-World APIs
>
> [4]. ZeroSearch: Incentivize the Search Capability of LLMs without Searching
>
> [5]. Agent Learning via Early Experience
>
> [6]. Scaling Agents via Continual Pre-training
>
>
> **We sincerely appreciate your thoughtful review and the valuable feedback you have provided. However, we believe that a score as low as 2 may merit some reconsideration in light of the technical clarification. If there are further concerns or questions, we are fully committed to addressing them comprehensively.**

---

> > ### Comment · Reviewer_t5Y1 · 2025-11-24
> >
> > Thanks for the response.
> >
> > > Weakness1: Seed-OSS-36B Performance Comparison
> >
> > **I believe that the statement in my review is not based on a factual misunderstanding of the model sizes involved.** When I pointed out that Seed-OSS-36B has similar model size with AgentScaler-30B-A3B, the authors argued that this is a factual misunderstanding of the model sizes. However, I believe that, when not stated explicitly, people typically refer to the actual total parameter count, instead of the number of activated parameters.
> >
> > Furthermore, if considering only activated parameters (as the authors suggested), then the model AgentScaler-30B-A3B should be comparable to GPT-OSS-120B-A5B, yet GPT-OSS-120B-A5B (without task specific finetuning) outperformed AgentScaler-30B-A3B on ACEBench-en.
> >
> > > Weakness2 & Question1: Performance Drop on ACEBench-zh
> >
> > The authors responded that ACEBench-zh is OOD, yet I don't think simply "it's OOD" could be a complete answer to my question. Particularly, the authors claimed that their method and data could enable transfer and generalization in section 5, and they placed Table 2 (eval results on ACEBench-zh) as an evidence to their claim. Then, ACEBench-zh being an OOD evaluation should not be a reason/excuse to why performance dropped on ACEBench-zh after training; rather, **it might indicate that the model overfits to the training data and could not generalize to OOD tasks reasonably well.** Therefore, I believe that this compromises the validity of the authors' claims of generalizability.
> >
> > > Weakness3: Motivation of Simulated Environments
> >
> > Regarding the authors' response, I agree that real environments suffer from several practical limitations. However, many of the limitations such as rate limits could be solved through sandboxing the environments (e.g. WebArena [1]). Also importantly, models outside of benchmarks need to operate with the real online environments where instability etc. might occur, and thus it is essential for models to learn to handle these. **Therefore, the authors didn't fully address my concerns whether these simulated environments could actually reflect the distribution of interaction in the real world environments.**
> >
> > [1] Zhou, Shuyan, et al. "WebArena: A Realistic Web Environment for Building Autonomous Agents." The Twelfth International Conference on Learning Representations.

---

> > > ### Author Response · Authors · 2025-11-28
> > > **Weakness 2 & Question 1: Performance Drop on ACEBench-zh**
> > >
> > > > **Weakness 2 & Question 1: Performance Drop on ACEBench-zh**
> > >
> > > We would like to further clarify more details of *out-of-distribution (OOD)* in our setting and explain why the observed behavior does *not* contradict our claims of generalization.
> > >
> > > - ACEBench-en is already OOD for our system.
> > > Our simulated environments are constructed using APIs from **ToolBench** and **API-Gen**, which **do not overlap** with the tool domains or schema structures in ACEBench-en. Therefore, ACEBench-en can evaluate **domain-level generalization to unseen tools**, rather than in-domain performance.
> > >
> > > - Tool-calling format is also OOD.
> > > Training is conducted using the `Qwen3-Hermes` tool-calling format, while  ACEBench adopts its `own custom parsing and execution protocol`. Strong performance on ACEBench-en therefore demonstrates **format-level generalization**, indicating that our model does *not* overfit to a single schema or interaction format.
> > >
> > > - ACEBench-zh further introduces cross-lingual OOD.
> > > Thus, ACEBench-zh simultaneously evaluates **domain-level, format-level, and language-level generalization**, making it a significantly more challenging benchmark.
> > >
> > > More specifically, the performance drop on the `special` subset of ACEBench-zh mainly arises from a high proportion of the following types:
> > >
> > > * **Lose Param**: Required parameters are missing in the query.
> > > * **Error Param**: Parameters violate format or constraint requirements.
> > > * **Exceed Function Ability**: The instruction exceeds the functional capability of all candidate tools.
> > >
> > > We did not explicitly model these instruction cases during training. Our current data synthesis focused on correct function execution flows. The model attempts to solve the user's problem (helpfulness) rather than rejecting it (refusal), leading to lower scores on these cases. This is a behavior alignment choice, not a lack of generalization capability.
> > > Despite this, the performance drop is limited to `specific` subsets, while `overall` ACEBench-zh scores still improve, indicating that our system achieves **net gains in domain- and format-level OOD generalization**, not merely cross-lingual robustness.
> > >
> > > We agree that **agent OOD generalization is a highly challenging and important problem**, and we greatly welcome continued discussion on this topic.

---

> > > ### Author Response · Authors · 2025-11-28
> > > **Weakness3: Motivation of Simulated Environments**
> > >
> > > > **Weakness 3: Motivation of Simulated Environments**
> > >
> > > We fully agree that real-world environments are ultimately the deployment target. However, from a *practical training perspective*, interacting with real APIs at scale presents several fundamental obstacles beyond rate limits, including:
> > >
> > > * **High financial cost**
> > > * **Strict permission and authentication requirements**
> > > * **Regional access restrictions**
> > > * **Unstable service availability**
> > >
> > > These constraints make it **extremely difficult to perform large-scale SFT/RL training** using real online APIs in a reproducible and controllable manner.
> > > The core motivation of our work, and the central conceptual message we aim to convey to the community, is that tool use can be fundamentally formulated as read/write operations over an underlying environment viewed as a database. Under this abstraction, the construction of large-scale simulated environments becomes not only feasible, but also an effective and scalable approach for enabling broad tool-use learning.
> > >
> > > Regarding robustness to tool-call failures, we explicitly **formalize each tool in Python code with exception handling**, and our SFT dataset includes **diverse error patterns**, such as invalid parameters and runtime failures, to better approximate real-world noise and instability.
> > >
> > > We acknowledge that simulated environments cannot perfectly replicate the full distribution of real-world interactions. However, we view them as a **necessary and controlled approximation** that enables *scalable, reproducible, and safe training*, while still preserving key challenges in tool invocation and error handling. We see this as a complementary step toward real-world deployment, rather than a replacement.
> > >
> > >
> > > **Once again, we sincerely thank the reviewer for the thoughtful and rigorous feedback. We believe this discussion has helped clarify several important aspects of our work, and we are happy to continue the exchange.**

---

> ### Author Response · Authors · 2025-11-28
> **Weakness 1: Seed-OSS-36B Performance Comparison**
>
> Thank you very much for your detailed follow-up and for the opportunity to further clarify these important points. We are grateful for the constructive discussion and respond to each concern below.
>
> > **Weakness 1: Seed-OSS-36B Performance Comparison**
>
> We believe that the disagreement here mainly stems from a difference in the interpretation of what constitutes “similar model size.” With the increasing adoption of MoE architectures due to their efficiency, the community now commonly encounters ambiguity between *total parameters* and *activated parameters* (e.g., GPT-OSS-120B-A5B, Kimi-K2-1T-A32B). We acknowledge that comparing MoE and Dense models is inherently challenging due to the discrepancy between active and total parameters.
>
> However, **Seed-OSS-36B is a fully dense model**, whereas our model is AgentScaler-30B-A3B, an MoE model. Under **both interpretations**, activated parameters and total parameters, our model is strictly **smaller** than Seed-OSS-36B. Therefore, we respectfully clarify that our original claim was not intended to rely on a favorable interpretation of parameter counting, but rather reflects this fundamental architectural and scale difference.
>
> From a research and technical contribution standpoint, we focus on **architecture- and size-comparable baselines**, namely Qwen3-Thinking-30B-A3B and Qwen3-Thinking-235B-A22B, where both architecture and effective capacity are aligned. On these controlled baselines, our method consistently achieves **substantial improvements**, as shown below:
>
> | Model                    | tau-Retail   | tau-Airline  | tau$^2$-Retail   | tau$^2$-Airline  | tau$^2$-Telecom  | AceBench |
> | ------------------------ | -------- | -------- | -------- | -------- | -------- | ------- |
> | Qwen3-Thinking-235B-A22B | 67.8     | 46.0     | 71.9     | 58.0     | 45.6     | 70.2    |
> | Qwen3-Thinking-30B-A3B   | 67.8     | 48.0     | 58.8     | 58.0     | 26.3        | 67.2    |
> | **AgentScaler-30B-A3B**  | **70.4** | **54.0** | **70.2** | **60.0** | **55.3** | **75.7**    |
>
> We additionally observe consistent gains when switching backbones to Qwen3-Thinking-4B and Qwen3-8B, further validating the **effectiveness and robustness of our method** across model scales.
>
> | Model                    | tau-Retail   | tau-Airline  | tau$^2$-Retail   | tau$^2$-Airline  | tau$^2$-Telecom  | AceBench |
> | ------------------------ | -------- | -------- | -------- | -------- | -------- | ------- |
> | Qwen3-Thinking-4B | 59.1     | 52.5   | 56.1     | 52.0     | 28.7     | 49.5  |
> | **AgentScaler-4B**   | **64.3**     | **54.0**   | **62.3**   |**56.0**     | **48.2**     | **65.9**    |
> | Qwen3-8B  | 45.2 | 25.0 | 41.2 | 30.5 |  23.5 | 65.9   |
> | **AgentScaler-8B**  | **50.4** | **42.0** | **58.8** | **44.0** | **45.4** | **67.4**    |
>
> **We sincerely appreciate this comment, which motivated us to refine the wording of our claims regarding model comparison to avoid ambiguity.**

---

### Comment · Area_Chair_iWkx · 2025-11-28
**Reminder: Engage with Authors During Rebuttal**

Dear reviewers, please engage with the authors during the rebuttal if you haven’t yet. The deadline is approaching; add clarifications and follow-ups in the submission thread to ensure a fair, informed decision. Thank you

---

### Note · Authors · 2026-01-06

I have read and agree with the venue's withdrawal policy on behalf of myself and my co-authors.